# Association of Spousal Social Support in Child-Rearing and Marital Satisfaction with Subjective Well-Being among Fathers and Mothers

**DOI:** 10.3390/bs14020106

**Published:** 2024-01-31

**Authors:** Hajime Iwasa, Yuko Yoshida, Kayoko Ishii

**Affiliations:** 1Department of Public Health, School of Medicine, Fukushima Medical University, Fukushima 960-1295, Japan; 2Tokyo Metropolitan Institute for Geriatrics and Gerontology, Tokyo 173-0015, Japan; 3Department of Midwifery and Maternal Nursing, School of Nursing, Fukushima Medical University, Fukushima 960-1295, Japan; kayokoi@fmu.ac.jp

**Keywords:** fathers in childcare, marital satisfaction, social support from spouse, subjective well-being

## Abstract

This study explored the association of spousal support and marital satisfaction with the subjective well-being of fathers and mothers using a mediation analysis. Data were gathered from 360 fathers and 338 mothers (aged 25–50 years). Subjective well-being was measured as an outcome using the Japanese version of the World Health Organization-Five Well-Being Index. Marital satisfaction was measured as a mediating variable using the Japanese version of the Marital Relationship Satisfaction Scale. Spousal social support (including instrumental, emotional, and appraisal support) was measured as an independent variable using four-point scales. Control variables were the father’s and mother’s ages, number of children, age of the youngest child, children going to nursery school or kindergarten, use of childcare services, self-evaluated low economic status, and weekday working hours. Among fathers, instrumental and emotional support had significant direct and indirect effects, with the latter mediated by the impact of marital satisfaction on subjective well-being; appraisal support had only significant indirect effects. Among mothers, instrumental support had significant direct and indirect effects; emotional and appraisal support had only significant indirect effects. Our findings indicate that social support from spouses has protective direct and indirect effects on subjective well-being among parents and suggest the need for mutual support between spouses to facilitate effective co-parenting.

## 1. Introduction

Childcare is a physically and mentally overburdening task. Fathers and mothers may develop physical and mental health problems due to the overload of childcare chores and the constant need to deal with the demands of their children [1,2,3]. However, mothers typically spend more time than fathers on housework and childcare. This trend is stronger in Japan than in the U.S. and Europe, with mothers spending considerably more time than fathers on housework and childcare [4].

Surely, the responsibility for child-rearing should be shouldered equally by fathers and mothers. In recent years, measures aimed at promoting fathers’ involvement in child-rearing, such as the Ministry of Health, Labor and Welfare’s “Ikumen (hunky dads) Project”, have been gaining national momentum in Japan [5]. One of the goals of the Healthy Parents and Children 21 (Tier 2) campaign is to encourage fathers to participate in child-rearing (Fundamental Issue C: “Percentage of fathers actively involved in child-rearing” ([target value 55%]) [6]. Increasing fathers’ participation in child-rearing reduces the burden on mothers and helps maintain their mental health [7,8,9]. Furthermore, it may positively affect child health and development (e.g., by preventing injury and obesity) [8,9].

However, promoting paternal parenting poses challenges. The length of working hours is reportedly an important determinant of childcare involvement [10,11], and increasing fathers’ childcare time is difficult while they are working long hours [12]. If, in addition to their current work conditions, Japanese fathers are forced to shoulder the burden of childcare, they may experience health problems. Furthermore, childcare anxiety in mothers and problems of child abuse may develop owing to fathers’ increased childcare burden [8,9]. Indeed, recent studies reported that fathers are also more likely to experience worsening mental health after the birth of their children [11,13]. Measures to support mothers in preventing mental health problems during child rearing are essential and are being expanded in Japan’s maternal and child health care system (e.g., free screening for postpartum depression during postnatal checkups). Meanwhile, in Japan, the implementation of support measures for fathers has been slower than for mothers. Research findings that contribute to effective support measures for fathers should inform further efforts to improve the understanding of the current status of fatherhood.

Therefore, effectively facilitating fathers’ participation in childcare requires investigating factors associated with paternal involvement in childcare and changeable factors that protect fathers’ mental health. Thus far, several factors associated with the mental health of fathers involved in child-rearing have been identified. Among personal factors, stress [13] and a history of mental illness [13,14] have been reported. Among domestic factors, younger child age [11,15], family structure (e.g., single parent) [15,16], satisfaction with the marital relationship [17,18,19], social support [13,19], and maternal depression [19] have been reported. Socioeconomic factors reported in previous studies include working hours [11], unemployment [14], high expenditure [11], and economic insecurity [19]. 

Domestic factors as determinants of fathers’ mental health are essential for the promotion of smooth co-parenting by couples [20]. Therefore, we focused on marital relationship satisfaction and social support from spouses as protective factors for paternal mental health. Satisfaction with the marital relationship refers to the degree of subjective satisfaction with the couple’s emotional ties [18]. It is an important source of solidarity while co-parenting and managing a family [20]. Marital relationship satisfaction is related to the mental health of both parents raising children [17,18,19]. It can be measured using the Quality Marriage Index [21,22]. Social support represents the functional aspect of social relationships surrounding individuals and refers to all kinds of support and assistance exchanged with others [23]. According to a classification in a previous study [23], the components of social support are emotional support, appraisal support, informational support, and instrumental support. Emotional support provides empathy and affection through respect, trust, concern, and listening. Appraisal support provides positive evaluation, such as affirmation, feedback, and social comparison. Informational support provides information necessary to solve problems, such as advice, suggestions, and instructions. Instrumental support includes the provision of tangible goods and services, such as material goods, money, labor, assistance through the donation of time, and environmental changes. Informational support can be considered a type of instrumental support.

As noted above, spousal social support and marital relationship satisfaction are factors that protect fathers’ mental health [13,17,18,19]. Social support has been reported to be a predictor of marital relationship satisfaction [17,24]. Furthermore, a study conducted on mothers found that social support from spouses (i.e., fathers) positively impacted marital relationship satisfaction, which resulted in better mental health of mothers [25]. Thus, we aimed to investigate whether marital satisfaction mediates the association between social support and mental health is observable in fathers. Whether the above mediating mechanisms differ by the type of social support from the spouse (i.e., instrumental, emotional, or appraisal support) also remains unknown.

The current study conducted a survey of fathers and mothers involved in child-rearing to examine the (i) conditions of subjective well-being reflecting the positive aspects of mental health and (ii) mediation effects of marital satisfaction on the relationship between social support from the spouse and subjective well-being of fathers when compared to mothers. The present study thus contributes to the maintenance of subjective well-being among fathers, as well as improved and effective involvement of fathers in childcare.

## 2. Materials and Methods

### 2.1. Participants

The survey was conducted in April 2020. We collaborated with an Internet research company with 1.2 million registered members to administer the survey. The company selected fathers based on stratification by the employment status of their wives (three categories: full-time employees, part-time employees, and unemployed), age (two categories: 25–35 and 36–50 years), and age of the youngest child (two categories: 0–3 and 4–6 years). Additionally, the company selected mothers based on stratification by their own employment status (three categories: full-time employees, part-time employees, and unemployed), age (two categories: 25–35 and 36–50 years), and age of the youngest child (two categories: 0–3 and 4–6 years). Recruitment included fathers and mothers who were married, whose youngest child was aged under 6 years, and who were not on maternity or childcare leave. Only fathers and mothers who were married and lived with their spouses were included. Among full-time employees, those who were working under the short working hour system due to child-raring were also included. Self-employed and freelance workers were not included. 

This study was conducted on fathers and mothers raising young children (under 6 years old) for the following reasons. The burden of raising young children (aged under 6 years) is considered to be greater than that of raising children of school age. It is also known that a couple forms a cooperative parenting style when the child is relatively young, and this significantly impacts the couple’s marital relationship later in life (e.g., “postpartum crisis”) [26]. Therefore, it is worthwhile to investigate individuals who need more support. Additionally, nowadays, the number of dual-earner households in Japan is increasing compared to previous years, and about half the households are dual-earners. Therefore, data on full-time employees, part-time employees, and unemployed mothers should be included in the analysis. Because the survey utilized a volunteer-based participation style rather than random sampling, the possibility of bias was considered in the mothers’ occupations. Therefore, when recruiting mothers, we selected an equal number of mothers from each of the three groups: full-time employees, part-time employees, and unemployed mothers.

### 2.2. Measurements

#### 2.2.1. Subjective Well-Being

Subjective well-being was measured using the Japanese version of the World Health Organization-Five Well-Being Index (WHO-5-J) [27], a self-administered questionnaire comprising five items assessing the degree of subjective well-being during the past two weeks on a six-point Likert scale, ranging from 0 (not at all) to 5 (all the time). Item scores were summed to obtain a total score (range: 0 to 25), with higher scores reflecting a higher level of subjective well-being (Cronbach’s alpha coefficients in this study were 0.879 and 0.867 for fathers and mothers, respectively).

#### 2.2.2. Marital Relationship Satisfaction

Satisfaction with the marital relationship was measured using the six-item Japanese version of the Marital Relationship Satisfaction Scale (e.g., “We have a flawless married life.”) [21,22]. Participants were asked to respond to each item on a four-point Likert scale, ranging from 1 (strongly disagree) to 4 (strongly agree). Scores for each item were added to obtain the total marital relationship satisfaction score. The higher the score, the more likely the participants were satisfied with their marital relationship (Cronbach’s alpha coefficients in this study were 0.930 and 0.940 for fathers and mothers, respectively).

#### 2.2.3. Spousal Social Support

We measured the social support in child-rearing that fathers and mothers receive from their spouses. Items for spousal social support were created and measured while referring to a previous study [28], with instrumental support defined as “being taught about how to do childcare and household chores”, emotional support as “listening to my problems and concerns in daily life”, and appraisal support as “receiving praise for involvement in childcare and household chores”. The participants were asked to respond to each item on a four-point Likert scale, ranging from 1 (strongly disagree) to 4 (strongly agree). The higher the scores, the more likely the participants were to receive each type of social support from spouses.

#### 2.2.4. Other Measurements

Parents’ age, number of children, age of the youngest child, working hours on weekdays, self-evaluation of low economic status, children attending nursery school or kindergarten, use of childcare services, childcare and housework hours on weekdays and holidays, leisure time on weekdays and holidays, and sleeping hours on weekdays and holidays were all assessed and used as control variables in the mediation analysis or in the description of participants’ basic characteristics. Parents’ ages were obtained from information held by the survey company. 

The participants were divided into two groups according to whether they worked an average of 12 h or more per day, considered “long working hours”. The use of nursery schools and kindergartens was dichotomized into “use” if the respondent used a licensed nursery school, unlicensed nursery school, kindergarten, or certified childcare center (*nintei-kodomo-en*) and “no use” if the respondent did not use any of these. Use of childcare support services was dichotomized as “use” if the respondent used childcare support services provided by childcare centers and local governments, babysitters, housework services, or other childcare related services and “no use” if the respondent did not use any of these services. Subjective economic status was evaluated through five classes (excellent, good, normal, poor, and very poor) and dichotomized into two categories (high economic status if they were in the former three classes and low economic status if they were in the latter two classes). 

The childcare environment is highly individualized, and the burden of childcare differs depending on the situation in which fathers and mothers are embedded (e.g., age of the children, use of childcare services, and economic status of the family), and it is conceivable that the nature of factors related to the mental health of fathers and mothers also differs. Therefore, this study used the above-mentioned control variables in the analytical model, with reference to previous studies [11,15,16,17,18,19].

### 2.3. Statistical Analysis

First, SPSS Statistics version 25 (IBM Corp., Armonk, NY, USA) was used for analysis. A *t*-test for continuous variables and a chi-squared test for categorical variables were conducted to compare the basic characteristics of fathers and mothers. A *t*-test for principal measurements was conducted for fathers and mothers, and Cohen’s d was estimated as an indicator of effect size. According to the guideline for effect size [29], Cohen’s d = 0.2, 0.5, and 0.8 are interpreted as small, medium, and large effects, respectively. 

Second, the SPSS macro PROCESS program (Model 4) [30] was used to perform the mediation analysis for fathers and mothers separately. A model was run using social support from the spouse (instrumental support, emotional support, and appraisal support) as the independent variable, subjective well-being as the dependent variable, marital satisfaction as the mediating variable, and the control variables mentioned above. Analysis was carried out by type of social support and gender. The significance level for all tests was set at 5%. The confidence intervals (CI) for the bootstrap method (estimated using 3000 samples) were set at 95% while estimating the significance of the indirect effects. We calculated the variance inflation factor to check for multicollinearity before performing the mediation analysis. We performed the mediation analyses according to Baron and Kenny’s framework (1986) [31] as follows. First, the statistical significance of the effect of social support from the spouse on subjective well-being was examined (i.e., total effect). Second, the statistical significance of the effect of social support on marital satisfaction was examined. Third, the statistical significance of the direct effect of social support on subjective well-being was examined. The direct effect is the value remaining after controlling for the mediation effect of marital satisfaction within the total effect. In addition to these three steps, the statistical significance of the indirect effect of marital satisfaction mediating the association between social support and subjective well-being was tested.

### 2.4. Ethical Considerations

This study was approved by the Ethics Committee of Fukushima Medical University (approval number: 2019-156; approval date: 13 September 2019). All participants were informed about the purpose of the study, that their participation was completely voluntary, that they could withdraw at any time without facing any penalty, and that no personally identifiable information would be gathered from the survey. All participants provided informed consent.

## 3. Results

Data for 360 fathers (mean age: 36.8 years; standard deviation: 5.5) and 338 mothers (mean age: 35.9 years; standard deviation: 4.9) were obtained. Table 1 shows the basic characteristics of participants. When comparing fathers’ and mothers’ basic characteristics, significant differences were found in age (fathers > mothers; *p* < 0.05), long working hours (fathers > mothers), children going to nursery school and kindergarten (fathers < mothers), time spent on childcare and housework on weekdays (fathers < mothers), time spent on childcare and housework on holidays (fathers < mothers), and leisure time on holidays (fathers > mothers). 

Descriptive statistics for social support, marital satisfaction, and subjective well-being are presented in Table 2. Fathers’ scores were significantly higher than mothers’ in instrument support (Cohen’s d = 1.19), emotional support (Cohen’s d = 0.19), appraisal support (Cohen’s d = 0.28), and marital satisfaction (Cohen’s d = 0.30). In the subjective well-being scores for fathers, the mean was 13.38, and the standard deviation was 5.47, while for mothers, the mean was 11.96, and the standard deviation was 5.28. After conducting a *t*-test to compare subjective well-being scores between fathers and mothers, statistically significant differences were found (Cohen’s d = 0.26). 

Regarding instrumental support, among fathers (Figure 1), the total effect of instrumental support on subjective well-being was significant (standardized regression coefficient [β] = 0.29); the effect of instrumental support on marital satisfaction was significant (β = 0.42); the direct effect of instrumental support on subjective well-being was significant (β = 0.18); and the indirect effect of marital satisfaction mediating the association between instrumental support and subjective well-being was significant (β = 0.12). Among mothers (Figure 1), the total effect of instrumental support on subjective well-being was significant (β = 0.35); the effect of instrumental support on marital satisfaction was significant (β = 0.37); the direct effect of instrumental support on subjective well-being was significant (β = 0.26); and the indirect effect of marital satisfaction mediating the association between instrumental support and subjective well-being was significant (β = 0.09).

Regarding emotional support, among fathers (Figure 2), the total effect of emotional support on subjective well-being was significant (β = 0.33); the effect of emotional support on marital satisfaction was significant (β = 0.54); the direct effect of emotional support on subjective well-being was significant (β = 0.20); and the indirect effect of marital satisfaction mediating the association between emotional support and subjective well-being was significant (β = 0.13). Among mothers (Figure 2), the total effect of emotional support on subjective well-being was significant (β = 0.25); the effect of emotional support on marital satisfaction was significant (β = 0.66); the direct effect of emotional support on subjective well-being was not significant; and the indirect effect of marital satisfaction mediating the association between emotional support and subjective well-being was significant (β = 0.21).

Regarding appraisal support, among fathers (Figure 3), the total effect of appraisal support on subjective well-being was significant (β = 0.25); the effect of appraisal support on marital satisfaction was significant (β = 0.49); the direct effect of appraisal support on subjective well-being was not significant; and indirect effect of marital satisfaction mediating the association between appraisal support and subjective well-being was significant (β = 0.15). Among mothers (Figure 2), the total effect of appraisal support on subjective well-being was significant (β = 0.20); the effect of appraisal support on marital satisfaction was significant (β = 0.57); the direct effect of appraisal support on subjective well-being was not significant; and the indirect effect of marital satisfaction mediating the association between appraisal support and subjective well-being was significant (β = 0.19).

## 4. Discussion

This study conducted a survey of fathers and mothers involved in child-rearing to examine the (i) conditions of subjective well-being and (ii) mediation effects of marital satisfaction on the relationship between social support from the spouse and subjective well-being among fathers, compared to mothers.

### 4.1. Differences in Participants’ Characteristics and Principal Measurements

A comparison of subjective well-being showed higher scores in fathers than mothers. A possible explanation for this may be that mothers are exposed to very long hours of childcare and housework [11]. The study also showed that mothers spent more time on childcare and housework on weekdays and holidays, while fathers spent more leisure time on weekends and holidays, suggesting that fathers’ participation in childcare should be encouraged to reduce the burden on mothers. The results also showed that fathers’ subjective well-being was lower than that in previous studies, such as the Danish general population studies (mean of 17.5) [32] and a study on a population of older adults in Japan (mean of 16.5 and 16.3 among men and women, respectively) [33]. Compared to those results, fathers involved in childcare in this study were under a reasonable amount of stress due to the combined burden of family and work. Therefore, fathers’ participation in childcare should be encouraged gradually and compassionately with consideration for their mental well-being, as well as measures for the mothers. As mentioned earlier, Japan’s Healthy Parents and Children 21 (Tier 2) campaign sets and promotes the goal of fathers’ participation in child-rearing [5]. However, a previous study claims that this is difficult to achieve because the current working hours leave little time for childcare and household chores [12]. It is important to think of childcare not as a burden on mothers alone, as was the case in the past, but as a responsibility of both the family and society. Various measures, including the enhancement of public and private childcare support services and improvement of the working environment, must be introduced to reduce this burden [34].

Fathers had higher values for emotional support, appraisal support, and marital satisfaction, but with small effect sizes. For instrumental support, fathers had greater values than mothers, with a large effect size. This outcome suggests that fathers receive more instrumental support from their wives during the child-rearing period for child-rearing and housework (i.e., fathers receive “on-the-job training” from their wives for child-rearing and housework) than do mothers from their husbands. While this is a positive point in terms of cooperative child-rearing, it may increase the burden on mothers, especially those who are physiologically prone to increased physical and mental fatigue in the early postpartum period [35,36,37]. Increasing opportunities for fathers to learn about childcare and housework before childbirth (e.g., by promoting parenting classes) [38] can encourage fathers’ participation in childcare, as well as reduce the burden on mothers.

### 4.2. Mediating Role of Marital Satisfaction on the Relationship between Social Support and Subjective Well-Being

For instrumental support, the results were similar for fathers and mothers. The direct effect was smaller than the total effect, but it was statistically significant, and the indirect effect mediated by marital satisfaction was also statistically significant, suggesting that marital satisfaction partially mediated the association between instrumental support and subjective well-being. Receiving instrumental support from one’s spouse improves mental health by reducing the daily burden (e.g., childcare and housework can be carried out more efficiently). The results also suggest that instrumental support from one’s spouse may increase marital satisfaction and ultimately improve subjective well-being. Receiving daily instrumental support from the spouse may increase the sense of trust between the couple, which, in turn, increases marital satisfaction [17,24], thereby improving the subjective well-being of both fathers and mothers.

The results were partially different for fathers and mothers regarding emotional support. For fathers, the direct effect was smaller than the total effect and statistically significant, and the indirect effect mediated by marital satisfaction was also statistically significant, suggesting that marital satisfaction partially mediated the relationship between emotional support and subjective well-being. Furthermore, for mothers, the total effect of emotional support was statistically significant, but the direct effect was not, suggesting that marital satisfaction completely mediated the association between emotional support and subjective well-being. In other words, emotional support may have little direct impact on subjective well-being among mothers; instead, high emotional support indirectly improves subjective well-being by improving marital satisfaction.

Similar results were obtained for fathers and mothers regarding appraisal support. Although the total effect of appraisal support was statistically significant, the direct effect was not, and the indirect effect mediated by marital satisfaction was also statistically significant, suggesting that marital satisfaction completely mediated the association between appraisal support and subjective well-being. This suggests that appraisal support may have little direct impact on subjective well-being; instead, high appraisal support indirectly improves subjective well-being through improved marital satisfaction.

### 4.3. Study Limitations

This study has a few limitations. First, the survey was conducted with registered members of an Internet survey company, which can be liable to “coverage error” or a discrepancy between the target population and the frame population used as a sampling method for participants [39]. Furthermore, because the survey was conducted online, participants in the current study were limited to those who could use the Internet on a daily basis, suggesting that the participants could be biased compared to the general population. 

Another limitation involves the causal relationships between variables. Because this study used a cross-sectional survey design, it was not possible to prove causal relationships. That is, it was not possible to determine from the present data whether receiving social support from one’s spouse led to better subjective well-being or, conversely, whether those with better subjective well-being were more likely to receive social support from their spouses. This is also true of models that included the mediating variable (i.e., marital satisfaction). In the future, a longitudinal survey design should be conducted to test the replicability and causality of these findings.

## 5. Conclusions

The present study explored the association of spousal support and marital satisfaction with the subjective well-being of fathers and mothers using a mediation analysis. Among fathers, instrumental and emotional support had significant direct and indirect effects, with the latter mediated by marital satisfaction’s impact on subjective well-being; appraisal support had only significant indirect effects. Among mothers, instrumental support had significant direct and indirect effects; emotional and appraisal support had only significant indirect effects. This indicates that spousal social support has both direct and indirect protective effects on subjective well-being among parents. Our findings suggest that mutual support between spouses is essential to improve co-parenting.

## Figures and Tables

**Figure 1 behavsci-14-00106-f001:**
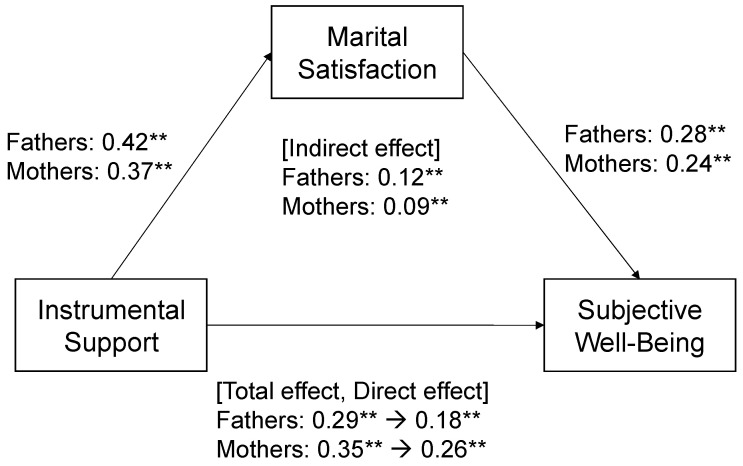
Mediation analysis was performed for fathers and mothers to explore the relationships between instrumental support and subjective well-being, mediated by marital satisfaction, adjusting for the father’s and mother’s age, number of children, age of the youngest child, children going to nursery school or kindergarten, use of childcare services, self-evaluated low economic status, and working hours on weekdays. Values indicate standardized regression coefficients. ** *p* < 0.01.

**Figure 2 behavsci-14-00106-f002:**
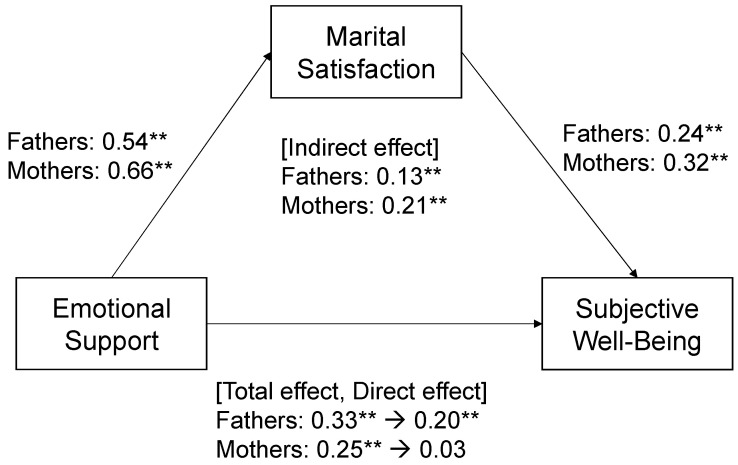
Mediation analysis was performed for fathers and mothers to explore the relationships between emotional support and subjective well-being, mediated by marital satisfaction, adjusting for the father’s and mother’s age, number of children, age of the youngest child, children going to nursery school or kindergarten, use of childcare services, self-evaluated low economic status, and working hours on weekdays. Values indicate standardized regression coefficients. ** *p* < 0.01.

**Figure 3 behavsci-14-00106-f003:**
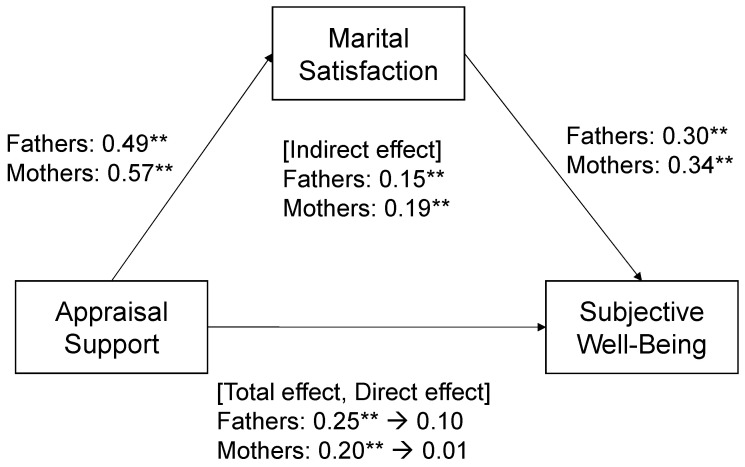
Mediation analysis was performed for fathers and mothers to explore the relationships between appraisal support and subjective well-being, mediated by marital satisfaction, adjusting for the father’s and mother’s age, number of children, age of the youngest child, children going to nursery school or kindergarten, use of childcare services, self-evaluated low economic status, and working hours on weekdays. Values indicate standardized regression coefficients. ** *p* < 0.01.

**Table 1 behavsci-14-00106-t001:** Basic characteristics of participants.

	Fathers (*n* = 360)	Mothers (*n* = 338)	*p* *
Age (years), mean (SD)	36.8 (5.5)	35.9 (4.9)	0.03
Number of children, mean (SD)	1.8 (0.7)	1.8 (0.8)	0.363
Age of the youngest child (years), mean (SD)	3.2 (1.8)	3.1 (1.7)	0.774
Working hours on weekdays (over 12 h), n (%)	32 (8.9)	6 (1.8)	<0.01
Self-evaluation of a low economic status n (%)	131 (36.4)	145 (42.9)	0.079
Children going to nursery school or kindergarten, n (%)	267 (74.2)	273 (80.8)	0.037
Use of childcare services, n (%)	31 (8.6)	29 (8.6)	0.998
Childcare and housekeeping hours on weekdays (hours), mean (SD)	2.0 (1.9)	7.9 (4.7)	<0.01
Childcare and housekeeping hours on holidays (hours), mean (SD)	5.5 (3.9)	10.8 (4.7)	<0.01
Leisure time on weekdays (hours), mean (SD)	2.2 (1.8)	1.9 (2.1)	0.103
Leisure time on holidays (hours), mean (SD)	4.1 (3.6)	2.9 (2.7)	<0.01
Sleep time on weekdays (hours), mean (SD)	6.7 (1.3)	6.7 (1.3)	0.357
Sleep time on holidays (hours), mean (SD)	7.5 (1.3)	7.3 (1.3)	0.066

Notes. * A *t*-test for continuous variables and a chi-squared test for categorical variables were conducted to compare fathers’ and mothers’ basic characteristics. SD = standard deviation.

**Table 2 behavsci-14-00106-t002:** Descriptive statistics for social support, marital satisfaction, and subjective well-being of fathers and mothers.

	Fathers (*n* = 360)	Mothers (*n* = 338)	*p* *	Effect Size **
Instrumental support, mean (SD)	2.75 (0.74)	1.80 (0.84)	<0.01	1.19
Emotional support, mean (SD)	2.62 (0.81)	2.45 (1.02)	<0.01	0.19
Appraisal support, mean (SD)	2.51 (0.79)	2.26 (0.97)	<0.01	0.28
Marital satisfaction, mean (SD)	17.41 (4.25)	16.03 (4.80)	<0.01	0.30
Subjective well-being, mean (SD)	13.38 (5.47)	11.96 (5.28)	<0.01	0.26

Notes. * A *t*-test for continuous variables was conducted to compare principal variables between fathers and mothers. ** Cohen’s d was estimated as an indicator of effect size. SD = standard deviation.

## Data Availability

Data that support the findings of this study cannot be publicly disclosed for ethical reasons. If you are a researcher who is interested in using these data, please request access to confidential data from the Fukushima Medical University Ethics Committee (rs@fmu.ac.jp).

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
