# Peer review of "Association of Spousal Social Support in Child-Rearing and Marital Satisfaction with Subjective Well-Being among Fathers and Mothers"

_behavsci, 2024, doi:10.3390/bs14020106_

Round 1

Reviewer 1 Report

Comments and Suggestions for Authors

Thank you for the opportunity to review your project. More research on fatherhood and coparenting is needed to normalize all parents being involved in their children's caregiving. I recommend this paper for acceptance after some minor revisions. 

One thing that I feel is missing from the paper is an explanation of why so many control variables were included in the analysis. Many of the control variables seem that they could play important roles in these conceptual models, so I would encourage the authors to provide their reasoning for why they were simply statistically controlled for rather than being included in the conceptual model. I am guessing it is due to statistical power, as not all could be included in one study.

In addition, I would like to encourage the authors to reframe this sentence from the Discussion: Therefore, encouraging fathers' 258 participation in childcare should be done in a gradual and compassionate manner with 259 consideration for their mental well-being.

Perhaps it is my own cultural lens affecting my interpretation of this sentence, but to me it sounds like the authors are suggesting that fathers need to be treated delicately to not scare them away from childcare. Who then do the authors suggest the responsibility lies with to make sure the child is cared for? If it is the mother, then what about the mother's mental well-being while she is being careful not to scare her male partner away from caregiving?

Author Response

Reviewer: 1

Thank you for reviewing our manuscript. We have revised the manuscript in accordance with the comments from the reviewer. The following are our responses to the reviewer’s comments. The revised parts are underlined in red.

Comments #1

Thank you for the opportunity to review your project. More research on fatherhood and coparenting is needed to normalize all parents being involved in their children's caregiving. I recommend this paper for acceptance after some minor revisions.

(Response #1)

Thank you for giving us the encouraging comment. In accordance with the suggestions from you, we made a revision.

Comments #2

One thing that I feel is missing from the paper is an explanation of why so many control variables were included in the analysis. Many of the control variables seem that they could play important roles in these conceptual models, so I would encourage the authors to provide their reasoning for why they were simply statistically controlled for rather than being included in the conceptual model. I am guessing it is due to statistical power, as not all could be included in one study.

(Response #2)

As per the comment, we added the explanation for including of control variables in the statistical models (line 165).

[The childcare environment is highly individualized, and the burden of childcare differs depending on the situation in which fathers and mothers are embedded (e.g., age of the children, use of childcare services, and economic status of the family), and it is conceivable that the nature of factors related to the mental health of fathers and mothers also differs. Therefore, this study used the above-mentioned control variables in the analytical model, with reference to previous studies (8, 12-16).]

Comments #3

In addition, I would like to encourage the authors to reframe this sentence from the Discussion: Therefore, encouraging fathers' 258 participation in childcare should be done in a gradual and compassionate manner with 259 consideration for their mental well-being.

Perhaps it is my own cultural lens affecting my interpretation of this sentence, but to me it sounds like the authors are suggesting that fathers need to be treated delicately to not scare them away from childcare. Who then do the authors suggest the responsibility lies with to make sure the child is cared for? If it is the mother, then what about the mother's mental well-being while she is being careful not to scare her male partner away from caregiving?

(Response #3)

We appreciate the reviewers' comments. Indeed, the responsibility for child rearing should be shouldered equally by both fathers and mothers. In addition, measures to support mothers in preventing mental health problems during child rearing are essential, and these are being expanded in Japan's maternal and child health care system (e.g., free screening for postpartum depression during postnatal checkups).

                On the other hand, it has been reported in recent years that fathers are also more likely to experience worsening mental health after the birth of their children (Takehara et al., 2020). In Japan, however, the implementation of support measures for fathers has been slower than for mothers. We believe that research findings that contribute to effective support measures for fathers should be further expanded and improved based on an understanding of the current status of fatherhood. The current study is based on such a perspective.

                Thus, we modified the sentence as follows (line 296):

[Therefore, fathers’ participation in childcare should be encouraged in a gradual and compassionate manner with consideration for their mental well-being, as well as measures for the mothers.]

In addition, in the text that follows, we stated that it is important to think of childcare not as a burden on mothers alone, as was the case in the past, but as a responsibility of the family and society together, as below (line 296).

[As mentioned earlier, Japan’s Healthy Parents and Children 21 (Tier 2) campaign sets and promotes the goal of fathers’ participation in child-rearing (1). However, a previous study claims that this is difficult to achieve because the current working hours leave little time for childcare and household chores (9). It is important to think of childcare not as a burden on mothers alone, as was the case in the past, but as a responsibility of both the family and society. Various measures, including the enhancement of public and private childcare support services and improvement of the working environment, must be introduced to reduce this burden (31)]

Reference

Takehara, K.; Suto, M.; Kato, T. Parental psychological distress in the postnatal period in Japan: a population-based analysis of a national cross-sectional survey. Sci. Rep. 2020, 10, 13770. doi:10.1038/s41598-020-70727-2.

Reviewer 2 Report

Comments and Suggestions for Authors

1. Regarding the title, I think the authors need to drop the last words: "among fathers". The content of this study is also largely about mothers. However, if the authors do not want that, they should explain why it is scientifically interesting to know marital satisfaction or subjective well being only among fathers. They should also explain why these phenomena are analysed in similar detail 'among mothers'.

2. In relation to the mediation analysis, my opinion is that we have quite a large number of variables, when social support is introduced into the analysis under the 4 forms of manifestation (emotional, appraisal, informational and instrumental). To this we add the stratification criteria (we are looking at differences between full-time employees, part-time employees, and unemployed; between two categories of age, to which the authors add two categories of youngest child). Over all the authors introduce direct effect and overall effect. This makes the study more difficult for the reader to understand. Therefore, to clarify the analysis, I propose a structured method to complete the sub-chapter "Discussion". I suggest the authors use Baron and Kenny's steps for mediation analysis - Baron, R. M., & Kenny, D. A. (1986). The moderator-mediator variable distinction in social psychological research: Conceptual, strategic, and statistical considerations. Journal of Personality and Social Psychology. The results can be present in the form of the three steps of mediation analysis.

3. I also believe that it should be stated why the authors chose this model of the relationship between the three variables (spouse social support, marital satisfaction and subjective well-being). Why is marital satisfaction the mediating variable and spouse social support is the independent variable? Why is it not the other way around? If subjective well-being is the dependent variable it means that the main aim of the study is the phenomenon of subjective well-being. In introduction the authors say that "The present study aims to contribute to the maintenance of fathers' mental health" (row 88).

4. I feel I must say that I think the comparative analysis between fathers and mothers is not a good idea. In the context of comparisons between men and women, the variables studied, but especially marital satisfaction, raise issues of social desirability. Such an approach fits well with the feminist perspective, although I am sure that was not the authors' intention. Of course, to change that comparative analysis a very large part of the article would have to be changed. That's why I don't press it.

5.   However, there remains one aspect that I think is important. The stratification criteria in the construction of the sample were presented but not explained. For example, I wonder why it was important that only parents with at least one child under the age of 6 were included in the sample. I also don't think that the differentiation between participants according to the three selection criteria related to place of work was explained.

Author Response

Reviewer: 2
Thank you for reviewing our manuscript. We have revised the manuscript in accordance with the comments from the reviewer. The following are our responses to the reviewer’s comments. The revised parts are underlined in red.

Comments #1
1. Regarding the title, I think the authors need to drop the last words: "among fathers". The content of this study is also largely about mothers. However, if the authors do not want that, they should explain why it is scientifically interesting to know marital satisfaction or subjective well being only among fathers. They should also explain why these phenomena are analysed in similar detail 'among mothers'.
(Response #1)
As per the comment, we modified the title of the manuscript as follows: 

[Association of social support from spouse and marital satisfaction with subjective well-being among fathers and mothers involved in childcare.]

Comments #2
2. In relation to the mediation analysis, my opinion is that we have quite a large number of variables, when social support is introduced into the analysis under the 4 forms of manifestation (emotional, appraisal, informational and instrumental). To this we add the stratification criteria (we are looking at differences between full-time employees, part-time employees, and unemployed; between two categories of age, to which the authors add two categories of youngest child). Over all the authors introduce direct effect and overall effect. This makes the study more difficult for the reader to understand. Therefore, to clarify the analysis, I propose a structured method to complete the sub-chapter "Discussion". I suggest the authors use Baron and Kenny's steps for mediation analysis - Baron, R. M., & Kenny, D. A. (1986). The moderator-mediator variable distinction in social psychological research: Conceptual, strategic, and statistical considerations. Journal of Personality and Social Psychology. The results can be present in the form of the three steps of mediation analysis.
(Response #2)
According to the reviewer comment, we added information regarding mediation analysis in the Method and Result section as below.

*In the Method section (line 187)
[We performed the mediation analyses according to Baron and Kenny’s framework (1986) (28) as follows. First, statistical significance of the effect of social support from the spouse on subjective well-being was examined (i.e., total effect). Second, statistical significance of the effect of social support on marital satisfaction was examined. Third, statistical significance of the direct effect of social support on subjective well-being was examined. The direct effect is the value remaining after controlling for the mediation effect of marital satisfaction within the total effect. In addition to these three steps, statistical significance of the indirect effect of marital satisfaction mediating the association between social support and subjective well-being was tested.]

*In the Result section (line 229).
[Regarding instrumental support, among fathers (Figure 1), the total effect of instrumental support on subjective well-being was significant (standardized regression coefficient [β] = 0.29); effect of instrumental support on marital satisfaction was significant (β = 0.42); direct effect of instrumental support on subjective well-being was significant (β = 0.18); and indirect effect of marital satisfaction mediating the association between instrumental support and subjective well-being was significant (β = 0.12). Among mothers (Figure 1), the total effect of instrumental support on subjective well-being was significant (β = 0.35); effect of instrumental support on marital satisfaction was significant (β = 0.37); direct effect of instrumental support on subjective well-being was significant (β = 0.26); and indirect effect of marital satisfaction mediating the association between instrumental support and subjective well-being was significant (β = 0.09).

Regarding emotional support, among fathers (Figure 2), the total effect of emotional support on subjective well-being was significant (β = 0.33); effect of emotional support on marital satisfaction was significant (β = 0.54); direct effect of emotional support on subjective well-being was significant (β = 0.20); and indirect effect of marital satisfaction mediating the association between emotional support and subjective well-being was significant (β = 0.13). Among mothers (Figure 2), the total effect of emotional support on subjective well-being was significant (β = 0.25); effect of emotional support on marital satisfaction were significant (β = 0.66); direct effect of emotional support on subjective well-being was not significant; and indirect effect of marital satisfaction mediating the association between emotional support and subjective well-being was significant (β = 0.21).

Regarding appraisal support, among fathers (Figure 3), the total effect of appraisal support on subjective well-being was significant (β = 0.25); effect of appraisal support on marital satisfaction was significant (β = 0.49); direct effect of appraisal support on subjective well-being was not significant; and indirect effect of marital satisfaction mediating the association between appraisal support and subjective well-being was significant (β = 0.15). Among mothers (Figure 2), the total effect of appraisal support on subjective well-being was significant (β = 0.20); effect of appraisal support on marital satisfaction was significant (β = 0.57); direct effect of appraisal support on subjective well-being was not significant; and indirect effect of marital satisfaction mediating the association between appraisal support and subjective well-being was significant (β = 0.19).]

Reference
Baron, R. M.; Kenny, D. A. The moderator-mediator variable distinction in social psychological research: Conceptual, strategic, and statistical considerations. Journal of Personality and Social Psychology 1986, 51, 1173-1182.

Comments #3
3. I also believe that it should be stated why the authors chose this model of the relationship between the three variables (spouse social support, marital satisfaction and subjective well-being). Why is marital satisfaction the mediating variable and spouse social support is the independent variable? Why is it not the other way around? If subjective well-being is the dependent variable it means that the main aim of the study is the phenomenon of subjective well-being. In introduction the authors say that "The present study aims to contribute to the maintenance of fathers' mental health" (row 88).
(Response #3)
Our study conducted a mediation analysis for the below reasons.

1. Previous studies reported that spousal social support and marital relationship satisfaction are factors that protect fathers' mental health (10, 14-16). 

2. Social support has been reported to be a predictor of marital relationship satisfaction (14, 21). Furthermore, in a study conducted on mothers, social support from spouses (i.e., fathers) had a positive impact on marital relationship satisfaction, which resulted in better mental health of mothers (22).

Thus, we aimed to examine a mediation effect of marital satisfaction on association between social support from spouses and mental we-being among fathers. In order to obtain useful information to improve mental health among fathers involved in childcare, it is necessary to know the protective factors of mental health, and its mechanisms underlying the association.

As per the latter suggestion from the reviewer, we address as follows. The present study used subjective well-being as a dependent variable. Thus, we modified the term as follows (line 87): 
[The present study thus contributes to the maintenance of subjective well-being among fathers, as well as improved and effective involvement of fathers in childcare.]

Comments #4
4. I feel I must say that I think the comparative analysis between fathers and mothers is not a good idea. In the context of comparisons between men and women, the variables studied, but especially marital satisfaction, raise issues of social desirability. Such an approach fits well with the feminist perspective, although I am sure that was not the authors' intention. Of course, to change that comparative analysis a very large part of the article would have to be changed. That's why I don't press it.
(Response #4)
Indeed, the results of this study compared fathers and mothers, showing that fathers worked longer hours than mothers and that mothers spent more time than fathers on childcare and housekeeping (Table 1).
    On the other hand, although fathers were higher than mothers in subjective well-being (Table 2), these values were lower than in the general population (Line 289). This suggests that support is needed not only for mothers but also for fathers.
    In Japan, there is currently less support for fathers than for mothers. it has been reported in recent years that fathers are also more likely to experience worsening mental health after the birth of their children (Takehara et al., 2020). In Japan, however, the implementation of support measures for fathers has been slower than for mothers. We believe that research findings that contribute to effective support measures for fathers should be further expanded and improved based on an understanding of the current status of fatherhood. The current study is based on such a perspective. 
    In addition, in the text that follows, we stated that it is important to think of childcare not as a burden on mothers alone, as was the case in the past, but as a responsibility of the family and society together, as below (line 296).

[As mentioned earlier, Japan’s Healthy Parents and Children 21 (Tier 2) campaign sets and promotes the goal of fathers’ participation in child-rearing (1). However, a previous study claims that this is difficult to achieve because the current working hours leave little time for childcare and household chores (9). It is important to think of childcare not as a burden on mothers alone, as was the case in the past, but as a responsibility of both the family and society. Various measures, including the enhancement of public and private childcare support services and improvement of the working environment, must be introduced to reduce this burden (31)]

Reference
Takehara, K.; Suto, M.; Kato, T. Parental psychological distress in the postnatal period in Japan: a population-based analysis of a national cross-sectional survey. Sci. Rep. 2020, 10, 13770. doi:10.1038/s41598-020-70727-2.

Comments #5
5.   However, there remains one aspect that I think is important. The stratification criteria in the construction of the sample were presented but not explained. For example, I wonder why it was important that only parents with at least one child under the age of 6 were included in the sample. I also don't think that the differentiation between participants according to the three selection criteria related to place of work was explained.
(Response #5)
As per the comment, we added the below information to the Method section (line 105).

Regarding stratification related to age group.
[This study was conducted on fathers and mothers who raise young children (under 6 years old) for the following reasons. The burden of raising young children (aged under six years) is considered to be greater than that of raising children of school age. It is also known that a couple forms a cooperative parenting style when the child is relatively young, and this significantly impacts the couple’s marital relationship later in life (e.g., “postpartum crisis”) (23). Therefore, it is worthwhile to investigate individuals who need more support.]

Additionally, regarding stratification related to place of work among mothers.
[Additionally, nowadays, the number of dual-earner households in Japan is increasing compared to previous years, and about half the households are dual-earners. Therefore, data on full-time employees, part-time employees, and unemployed mothers should be included in the analysis. Because the survey utilized a volunteer-based participation style rather than random sampling, the possibility of bias was considered in the mothers’ occupations. Therefore, when recruiting mothers, we selected an equal number of mothers from each of the three groups: full-time employees, part-time employees, and unemployed mothers.]

Reference
Uchida, S.; Tsuboi, K. Postpartum crisis. Poplar Publishing Co. 2013.
